# A Shortcut from Metabolic-Associated Fatty Liver Disease (MAFLD) to Hepatocellular Carcinoma (HCC): c-MYC a Promising Target for Preventative Strategies and Individualized Therapy

**DOI:** 10.3390/cancers14010192

**Published:** 2021-12-31

**Authors:** Feifei Guo, Olga Estévez-Vázquez, Raquel Benedé-Ubieto, Douglas Maya-Miles, Kang Zheng, Rocío Gallego-Durán, Ángela Rojas, Javier Ampuero, Manuel Romero-Gómez, Kaye Philip, Isioma U. Egbuniwe, Chaobo Chen, Jorge Simon, Teresa C. Delgado, María Luz Martínez-Chantar, Jie Sun, Johanna Reissing, Tony Bruns, Arantza Lamas-Paz, Manuel Gómez del Moral, Marius Maximilian Woitok, Javier Vaquero, José R. Regueiro, Christian Liedtke, Christian Trautwein, Rafael Bañares, Francisco Javier Cubero, Yulia A. Nevzorova

**Affiliations:** 1Department of Immunology, Ophthalmology and ENT, School of Medicine, Complutense University of Madrid, 12 de Octubre (imas12) Health Research Institute, 28040 Madrid, Spain; feiguo@ucm.es (F.G.); olgaeste@ucm.es (O.E.-V.); rabenede@ucm.es (R.B.-U.); kzheng@ucm.es (K.Z.); chaochen@ucm.es (C.C.); arlamas@ucm.es (A.L.-P.); regueiro@med.ucm.es (J.R.R.); Rafael.banares@salud.madrid.org (R.B.); fcubero@ucm.es (F.J.C.); 2Department of Obstetrics and Gynaecology, The Affiliated Drum Tower Hospital of Nanjing University Medical School, Nanjing 210023, China; 3Department of Physiology, Genetics and Microbiology, Faculty of Biology, Complutense University Madrid, 28040 Madrid, Spain; 4Institute of Biomedicine of Seville (IBiS), SeLiver Group, Virgen del Rocío University Hospital/CSIC/University of Seville, 41013 Seville, Spain; dmaya-ibis@us.es (D.M.-M.); rgallego-ibis@us.es (R.G.-D.); marojas-ibis@us.es (Á.R.); jampuero-ibis@us.es (J.A.); mromerogomez@us.es (M.R.-G.); 5UCM Digestive Diseases, Virgen del Rocío University Hospital, 41013 Seville, Spain; 6Centro de Investigación Biomédica en Red de Enfermedades Hepáticas y Digestivas (CIBEREHD), 28220 Madrid, Spain; jsimon.ciberehd@cicbiogune.es (J.S.); mlmartinez@cicbiogune.es (M.L.M.-C.); j.vaquero@iisgm.com (J.V.); 7Department of Anesthesiology, Zhongda Hospital, School of Medicine, Southeast University, Nanjing 210009, China; sunjie@seu.edu.cn; 8Department of Medicine, University of Seville, 41009 Seville, Spain; 9Department of Pathology, Nottingham University Hospitals NHS Trust, Queen’s Medical Centre Campus, Nottingham NG7 2UH, UK; Philip.Kaye@nuh.nhs.uk (K.P.); Isioma.Egbuniwe@nottingham.ac.uk (I.U.E.); 10Department of General Surgery, Wuxi Xishan People’s Hospital, Wuxi 214000, China; 11Department of Hepatic-Biliary-Pancreatic Surgery, The Affiliated Drum Tower Hospital of Nanjing University Medical School, Nanjing 210023, China; 12Liver Disease Laboratory, Center for Cooperative Research in Biosciences (CIC bioGUNE), Basque Research and Technology Alliance (BRTA), 48160 Derio, Spain; tcardoso@cicbiogune.es; 13Department of Internal Medicine III, University Hospital RWTH Aachen, 52074 Aachen, Germany; joreissing@ukaachen.de (J.R.); tbruns@ukaachen.de (T.B.); M.Woitok@eclevar.com (M.M.W.); cliedtke@ukaachen.de (C.L.); ctrautwein@ukaachen.de (C.T.); 14Department of Cell Biology, Complutense University School of Medicine, 28040 Madrid, Spain; mgomezm@med.ucm.es; 15Servicio de Aparato Digestivo, Hospital General Universitario Gregorio Marañón, 28009 Madrid, Spain; 16Instituto de Investigación Sanitaria Gregorio Marañón (IiSGM), 28007 Madrid, Spain

**Keywords:** metabolic-associated fatty liver disease (MAFLD), c-myc, oncogene, tumorigenesis, metformin

## Abstract

**Simple Summary:**

Metabolic-associated fatty liver disease (MAFLD) is a chronic liver disease associated with obesity, diabetes mellitus type 2 (DM2), and hyperlipidemia. It can also progress to end-stage hepatocellular carcinoma (HCC); the underlying mechanisms are still unknown, but endogenous (i.e., genetic) factors such as oncogenes have been suggested to play a role. We found that c-MYC transgenic mice with ageing are prone to develop obesity, metabolic syndrome (MS), and abnormal accumulation of lipids in the liver compared to control mice. A short-term application of the Western diet (WD) significantly worsened the phenotype and accelerate HCC development. Importantly, we found that metformin as therapeutic approach significantly attenuated MAFLD phenotype in transgenic mice. We also observed that c-MYC is up-regulated in human patients with MAFLD and MAFLD-related HCC. Altogether the current study suggests an important role of the oncogene c-MYC during the progression from MAFLD to HCC and makes c-MYC a possible target for preventative strategies and individualized therapy.

**Abstract:**

Background: Metabolic-associated fatty liver disease (MAFLD) has risen as one of the leading etiologies for hepatocellular carcinoma (HCC). Oncogenes have been suggested to be responsible for the high risk of MAFLD-related HCC. We analyzed the impact of the proto-oncogene c-MYC in the development of human and murine MAFLD and MAFLD-associated HCC. Methods: alb-myc^tg^ mice were studied at baseline conditions and after administration of Western diet (WD) in comparison to WT littermates. c-MYC expression was analyzed in biopsies of patients with MAFLD and MAFLD-associated HCC by immunohistochemistry. Results: Mild obesity, spontaneous hyperlipidaemia, glucose intolerance and insulin resistance were characteristic of 36-week-old alb-myc^tg^ mice. Middle-aged alb-myc^tg^ exhibited liver steatosis and increased triglyceride content. Liver injury and inflammation were associated with elevated ALT, an upregulation of ER-stress response and increased ROS production, collagen deposition and compensatory proliferation. At 52 weeks, 20% of transgenic mice developed HCC. WD feeding exacerbated metabolic abnormalities, steatohepatitis, fibrogenesis and tumor prevalence. Therapeutic use of metformin partly attenuated the spontaneous MAFLD phenotype of alb-myc^tg^ mice. Importantly, upregulation and nuclear localization of c-MYC were characteristic of patients with MAFLD and MAFLD-related HCC. Conclusions: A novel function of c-MYC in MAFLD progression was identified opening new avenues for preventative strategies.

## 1. Introduction

Metabolic dysfunction-associated fatty liver disease (MAFLD) is a spectrum of chronic liver diseases characterized by excessive accumulation of triglycerides (TG) in hepatocytes. The condition ranges from isolated excessive hepatocyte TG accumulation and steatosis (fatty liver (FL)), to hepatic TG accumulation plus inflammation and hepatocyte injury (steatohepatitis (SH)) and finally to hepatic fibrosis and cirrhosis and/or end-stage hepatocellular carcinoma (HCC) [1].

Since its first description in the early 1980s, as “a poorly understood and hitherto unnamed liver disease”, MAFLD has considerably evolved and progressively gained recognition among clinical and research hepatologists [2,3]. Forty years later, the alarming epidemic of obesity and diabetes mellitus type 2 (DM2), has fueled an increasing prevalence of MAFLD and rendered it a societal health problem worldwide with a prevalence up to 24% worldwide. For example, by 2015, MAFLD cases in the US were estimated to affect 83.1 million people and predicted to increase over 100 million by 2030 [4]. In Spain, the total prevalence of MAFLD was estimated in 10.53 million cases in 2016 with a further potential to increase to 12.7 million by 2030 [1,5].

Patients with MAFLD commonly consume high amounts of processed food with high fructose levels and/or high fat content and tend to adopt a sedentary lifestyle with low levels of physical activity. However, besides these environmental or exogenous factors many other factors often determine the progression of MAFLD and the development of MAFLD-associated carcinogenesis. For instance, 42% of MAFLD patients develop advanced SH stages and only 2.4–12.4% finally develop HCC [6]. Altogether, the huge variation in the susceptibility to the disease progression clearly indicates that among risk factors, endogenous (i.e., genetic) factors are of major relevance. 

The identification of such risk factors, which synergistically with environmental factors (such as unhealthy eating habits) drive MAFLD in the direction of cancer development would have great benefits for optimization of therapeutic strategies. 

The proto-oncogene c-MYC is a transcription factor which plays a central role in multiple biological processes including cell proliferation, cell growth, energy metabolism and apoptosis. Genetic alterations of c-MYC expression have been found in approximately 70,000 U.S. cancer deaths per year, including HCC [7]. Alb-myc^tg^ mice with overexpression of c-MYC only in hepatocytes became the touchstones of experimental models of murine liver carcinogenesis due to high HCC predisposition [8].

However, the development of HCC is usually preceded by chronic liver injury and ongoing liver diseases. During the last decade, we systematically investigated the role of c-MYC in chronic liver disease (CLD). We showed that upregulation of c-MYC in hepatocytes either due to gene amplification or as a response to inflammatory liver injury results in hepatic stellate cells (HSCs) preactivation. Primed HSC have a higher potential to proliferate and produce extracellular matrix (ECM) especially after a second profibrotic hit [9]. One of these hits can be an excessive alcohol consumption. We found the synergistic harmful effect of increased c-MYC expression and alcohol consumption on alcohol-associated liver disease (ALD) progression in a multistep process of exacerbated ROS production, mitochondrial and ER dysfunction, cell death, impaired cell proliferation and collagen deposition [10].

Despite the crucial recent increase in MAFLD and MAFLD-associated cancer, the specific link between c-MYC and MAFLD has not yet been addressed. In the present study, we aimed to elucidate the role of c-MYC in the initiation, progression and complications of MAFLD.

## 2. Materials and Methods

### 2.1. Human Liver Samples

Liver samples for IHC analysis were obtained from the Andalusian Biobank (Ethics Committee of the Junta de Andalucía C.P. S2100062—C.I. 0458-N-21.) from patients who were subjected to a liver biopsy for diagnostic confirmation purposes or from patients receiving a liver transplant. Diagnosis of MAFLD includes an evaluation of clinical risk factors and a histological evaluation of tissue together with the exclusion of other causes of liver disease (alcohol use disorders or excessive alcohol consumption, chronic hepatitis C or B, autoimmune, other causes). Two of these six patients did not show an obvious accumulation of fat ≥ 5% despite the presence of cirrhosis in the liver and their classification relies on clinical parameters. HCC diagnosis was performed by an experiences pathologist based on histopathological criteria (trabecular-resembling and solid patterns). Samples from HCC patients were included an IHC analysis for adjacent nontumoral and tumoral tissue. Appendix A summarize the patients’ characteristics.

### 2.2. Maintenance and Treatment of Mice

We used 12-week-old male transgenic mice carrying c-MYC transgene under the control of the hepatocyte-specific albumin promoter (alb-myc^tg^) as previously described [11] and WT littermates as controls. Diets used for the development of MAFLD are shown in Appendix A.

All animal procedures were carried out according to Spanish legal requirements and animal protection law and approved by the authority of environment conservation and consumer protection of the regional government of Madrid (PROEX210/18 and PROEX125.1/20). All animals were maintained in the animal facility at the School of Biology, Complutense University of Madrid, in a temperature-controlled room with 12 h light/dark cycle with free access to food and water according to the guidelines of the Federation for Laboratory Animal Science Associations (FELASA).

### 2.3. Supplementary Material and Methods

See Appendix A for further information regarding material and methods on animal experimentation, glucose and insulin tolerance test (GTT), histological analysis, immunofluorescence (IF) and immunohistochemistry (IHC) staining, image analysis, RNA isolation and RT-qPCR, Western blot, and TG quantification.

### 2.4. Statistical Analysis

Data are expressed as mean ± standard deviation of the mean. Statistical significance was determined by two-way analysis of variance followed by a Student’s t test. *p* values for significance are indicated as follows: * *p* < 0.05; ** *p* < 0.01; *** *p* < 0.001; **** *p* < 0.0001.

## 3. Results

### 3.1. c-MYC Is Induced in Patients with MAFLD and MAFLD-Related HCC

First, we analyzed the possible genetic alterations of c-MYC in a comprehensive dataset from 353 HCC patients generated by the Cancer Genome Atlas Research Network (https://cancergenome.nih.gov/, accessed on 16 December 2021) [12]. Genetic alterations of c-MYC, mainly amplifications, were detected in 41 patients (12%) (Appendix A).

Next, we narrowed down the etiology of HCC and validated the expression of c-MYC in liver biopsies from a cohort of patients with clinically and histologically diagnosed MAFLD-related HCC (Appendix A). We observed a higher rate of c-MYC nuclear expression inside tumor nodules compared to corresponding tumor-adjacent areas (Figure 1A,B and Appendix A).

Finally, we determined the expression of c-MYC in liver samples from six patients with MAFLD and advanced liver fibrosis (F3–F4) (Appendix A). c-MYC protein expression was increased and deposition was localized mainly in nuclei of hepatic cells along fibrotic areas (Figure 1C and Appendix A). Overall, these data demonstrated that c-MYC expression is increased in MAFLD and MAFLD-related HCC.

### 3.2. Spontaneous Metabolic Syndrome Is a Feature of 36-Week-Old c-MYC Transgenic Mice

To better define the relevance of c-MYC activation in MAFLD, we took advantage of transgenic mice with overexpression of c-MYC in hepatocytes (alb-myc^tg^). Interestingly, alb-myc^tg^ mice developed rapid and significantly greater weight gain compared to control littermates on normal chow diet (Figure 2A) with similar caloric intake (Appendix A). These changes were obvious already in 10-week-old animals, and became more pronounced as the mice grew older, and finally resulted in robust increase in the body mass index (BMI) in alb-myc^tg^ mice at the experimental end-point (36 weeks) (Figure 2A,B). 

Accordingly, a significant increase of epididymal white adipose tissue (eWAT) in alb-myc^tg^ mice was apparent by MRI scan images (Appendix A) and further confirmed during necropsy (Figure 2C). Hence, H&E staining of eWAT in alb-myc^tg^ mice revealed significant adipocyte hypertrophy and numerous crown-like structures (CLS), formed by F4-80 positive macrophages aggregates (Figure 2D and Appendix A).

Moreover, we found a significant increase in total cholesterol and development of hypertriglyceridemia persisting in alb-myc^tg^ from 36 weeks of age (Figure 2E). Importantly, c-MYC transgenic mice developed hyperglycaemia after 6 h of fasting and demonstrate considerably impaired glucose tolerance in the glucose tolerance test (GTT) (Figure 2F and Appendix A). These changes were accompanied by significantly delayed glucose clearance in the insulin tolerance test (ITT) 30, 60, 90 min after insulin injection (Appendix A).

Altogether, 36-week-old alb-myc^tg^ mice were prone to spontaneously develop obesity, hypertriglyceridemia, hypercholesterolemia, glucose intolerance and insulin resistance (IR), characteristic of MS.

### 3.3. MAFLD-Associated Changes in the Liver of Middle-Aged alb-myc^tg^ Mice

Recent studies support a central role of MS in the pathogenesis of MAFLD [13]. We did not detect any significant differences in liver weight or in hepatosomatic ratio between transgenic and WT mice (Appendix A). However, careful examination of liver histology revealed that alb-myc^tg^ mice with ageing are much more prone to develop abnormal accumulation of lipids in the liver compared to control animals. H&E staining showed microvesicular steatosis characterized by distended hepatocytes with small lipid vesicles and foamy appearing cytoplasm (Figure 3A). These findings were confirmed by Oil-Red-O staining (Figure 3B,C) and by direct quantification of hepatic TG (Figure 3D).

Blinded quantitative analysis performed by an experienced pathologist revealed that middle-aged alb-myc^tg^ mice exhibited micro- and macrovesicular steatosis grade 1 associated with hepatocyte ballooning reaching in most of the animals a 2.4 NAFLD activity score (NAS) (Figure 3E).

Consistent with an abnormal lipid metabolism, the expression of *Scd1* gene increased significantly in hepatic tissue from 36-week-old alb-myc^tg^ (Appendix A).

Hence, hepatic accumulation of lipids in transgenic animals leads to compensatory enhancement of mitochondrial β-oxidation and increased expression of *Cpt1* and *Acadm* genes (Figure 4A). Obviously, this process is insufficient to normalize lipid levels and in turn generate considerable amounts of reactive oxygen species (ROS) and oxidative stress. Consequently, a lipid peroxidation marker, 4-HNE was increased in the liver parenchyma of alb-myc^tg^ mice (Figure 4B). Another important cause of ROS overproduction in alb-myc^tg^ is CYP2E1 overexpression (Figure 4C and Appendix A). Enhanced ROS production triggers the unfolded protein response (UPR) and compensatory *Ucp2* activation (Appendix A). Excess workload misfolded proteins cause activation of the UPR, presented in c-myc^tg^ mice by activation of CHOP (Figure 4C and Appendix A).

Concomitant with these data, transmission electron microscopy (TEM) revealed large and aggregated lipid deposition; disorganized and fragmented rough endoplasmic reticulum (ER) in c-MYC transgenic mice (Appendix A).

Prolonged activation of the UPR and ROS production in alb-myc^tg^ mice likely contributed to liver damage and caused a mild but significant increase of the plasma levels of ALT (Figure 4D), a major clinical indicator of cellular liver injury. Consistently, c-MYC overexpression induced moderate apoptotic cell death in livers of transgenic mice as evidenced by cleaved caspase 3 (CC3) immunoblot (Figure 4E and Appendix A). Death of hepatocytes triggers compensatory proliferation in surrounding cells to maintain tissue homeostasis (Figure 4E). The analysis of PCNA (Figure 4E and Appendix A) and Ki-67 (Appendix A) expression revealed moderate hepatic proliferation in the livers from alb-myc^tg^ mice. 

Hepatic steatosis and cell death in the liver further caused immune cell infiltration and hepatic inflammation [14]. Alb-myc^tg^ animals showed an increased accumulation of CD45- and F4/80-myeloid positive cells as assessed by IF staining (Figure 4F and Appendix A). Finally, analysis of liver fibrosis, the hallmark of SH, revealed marked fibrosis in livers from alb-myc^tg^ mice by SR staining (Figure 4G and Appendix A). 

Increased glutamine catabolism is a major metabolic feature of rapidly proliferating cells such as hepatocytes and αSMA-positive myofibroblasts. Indeed, c-MYC has been identified as an upstream regulator of GLS1 [15]. Therefore, we found increased expression of glutaminase 1 (GLS1) and glutamine synthetase (GS) in the livers of alb-myc^tg^ mice (Appendix A).

Finally, gene array data in the livers of alb-myc^tg^ mice under basal conditions revealed upregulation of pathways related to cytochrome P450 and ER-stress response, lipid/fatty acid metabolism, glucose homeostasis, DNA damage and proliferation/cell cycle, in accordance with previous publications [10] (Appendix A).

In agreement with previous reports [16], 20% of transgenic mice at the age of 52 weeks develop liver tumor (Appendix A).

In summary, our data strongly suggest an important role of the oncogene c-MYC in murine livers for MAFLD initiation and subsequent tumor promotion.

### 3.4. WD Feeding Accelerates Liver Injury, Steatohepatitis, Fibrosis and Initiation of MAFLD Associated HCC in alb-myc^tg^ Mice

Next, we fed our transgenic model a WD using 12-week-old alb-myc^tg^ and WT mice for 24 weeks. alb-myc^tg^ mice fed with WD did not gain weight faster (Appendix A), however; they developed higher BMI compared to WT mice at the experimental end-point (Figure 5A,B). This was accompanied by hypertrophy and inflammation of adipocytes and eWAT inflammation (Figure 5B and Appendix A). Additionally, transgenic mice showed significant upregulation of blood glucose after fasting, increase in total serum cholesterol and hypertriglyceridemia (Figure 5C). WD induced micro- and macrovesicular steatosis in both groups of mice. However, the level of steatosis was much stronger in transgenic mice compare to WT littermates (Figure 5D and Appendix A). This was further confirmed by measurement of hepatic TG (Figure 5D, right) and by pathological evaluation of NAFLD activity (Appendix A). Consistently, WD-fed alb-myc^tg^ mice 24 weeks of exhibited stronger hepatic inflammation (CD45 positive cells) (Figure 5E and Appendix A) and extensive hepatic fibrosis, assessed by SR staining (Figure 5F and Appendix A) and αSMA (Figure 5G, middle, Appendix A) expression. SH in the liver was accompanied by high compensatory proliferation, confirmed by elevated numbers of Ki-67 positive cells (Figure 5H, right, Appendix A) and apoptotic cell death, detected by CC3 overexpression (Figure 5I and Appendix A).

Finally, with the chronic feeding with a WD for a period of 40 weeks led to the development of multiple tumor foci in 100% alb-myc^tg^ mice (five out of five). In contrast, none of the WT animals fed with WD had visible tumor nodules at this time point (zero out of five) (Figure 6A).

H&E (Figure 6B) and SR (Figure 6C) of the liver demonstrated well demarcated nodules surrounded by extensive collagen deposition. Moreover, massive accumulation of highly proliferative Ki-67 positive cells was detected inside the nodules (Figure 6D). Importantly, these changes were accompanied by extensive c-MYC nuclear expression in the liver tissue of transgenic mice (Figure 6E).

Altogether, our data strongly support the hypothesis that oncogene c-MYC and WD synergistically drive MAFLD progression towards carcinogenesis.

### 3.5. Anti-MAFLD and MS Potential of Metformin in alb-myc^tg^ Mice

Metformin, an oral biguanide is now generally accepted as the first-line treatment for DM2. Moreover, metformin has demonstrated antineoplastic effects in several types of tumors, including HCC [17,18]. Since DM2 and IR play the key roles in the pathogenesis of MAFLD, we next evaluated the effects of metformin treatment in c-MYC transgenic mice.

Alb-myc^tg^ fed with chow diet enriched with 0.1% metformin gained significantly less weight (Appendix A) and exhibited significantly lower BMI (Figure 7A) at experimental end-point (36 weeks) in comparison with alb-myc^tg^ mice fed with chow diet. Importantly, metformin-enriched chow diet led to considerable decrease of fasting glycaemia as well as improved glucose tolerance in GTT (Figure 7B).

Transgenic mice fed with met+chow diet accumulated less adipose eWAT tissue (Figure 7C) and demonstrated improved parameters of lipidemia (TG and cholesterol in blood) (Figure 7D). Remarkably, metformin-enriched chow diet improved hepatic steatosis (Figure 7E,F), attenuated collagen accumulation (Figure 7G), decreased AST level (Appendix A) and mildly improved hepatic proliferation (Figure 7H and Appendix A) in transgenic animals. Importantly, these positive changes were not a consequence of a direct inhibition of c-MYC expression (Appendix A), but rather effects of diminished lipogenic sterol-regulated element-binding protein (SREBP1) overexpression (Figure 7I and Appendix A) in alb-myc^tg^ after metformin treatment.

Altogether, our data showed the efficacy of metformin on lipogenic targeting in c-MYC transgenic mice leading to positive histological and biochemical alterations.

## 4. Discussion

c-MYC, ‘‘the oncogene from hell’’, is associated with more than half of all human cancers [19]. Gains of chromosome 8q22-24 region, encoding c-MYC are amongst the earliest genomic events associated with liver cancer development [20]. According to the Cancer Genome Atlas Research Network c-MYC alterations are reported in up to 12% of HCC [12]. Novel data suggest that deregulation of c-MYC function is not only associated with HCC development, but also with CLD (e.g., ALD, viral hepatitis, liver fibrosis/cirrhosis, for hepatoblastoma and cholangiocarcinoma, etc.) [8].

Since defective-MYC expression is an early event in carcinogenesis, the characterization of the consequences of c-MYC overexpression in hepatocytes is of ultimate interest. Indeed, hepatocyte-specific alb-myc^tg^ mice are predisposed to HCC albeit a long period of latency [11,21]. 20% alb-myc^tg^ mice at the age of 52 weeks displayed liver tumors that were mostly mono- or two-nodular. The gene signature of these tumors exactly resembles slow growing human HCC [16,22].

Hence, the oncogene c-MYC, “super-transcription factor”, is not only a master regulator of cell growth and cell cycle arrest, but also plays an essential role in the regulation of metabolic reprogramming. Proliferating cells change their metabolic demands compared to quiescent cells as only global metabolic alterations permit tumor cells to survive and proliferate despite adverse conditions [23,24].

Our experiments revealed that middle-aged c-MYC transgenic mice on standard chow diet are prone to develop abnormal accumulation of lipids in the liver compared to control mice. Consistently, gene expression profile of alb-myc^tg^ revealed significant alterations in FA metabolism. Activation of lipid synthesis is very important for fast growing cells, because lipids are fundamental for functions such as membrane generation, protein modification and bioenergetics requirements. In a wide variety of tumors, de novo FA synthesis is activated independently of the levels of circulating lipids [25]. The main product of FA synthesis in the cytoplasm is saturated palmitic acid, formed by rate-limiting enzyme—SCD1—on the cytosolic side of the ER. Consequently, SCD was upregulated in alb-myc^tg^ livers. Hence, SCD1 overexpression has been reported as promotor in liver fibrosis and tumor development [26].

FA overload in c-MYC transgenic hepatocytes acted as both a substrate and an inducer of microsomal cytochrome P-450 (CYP2E1) and fatty acid oxidation systems (Cpt1, Acadm) that generate reactive oxygen species resulting in oxidative stress (20). Lipid peroxidation products as well as proteins modified by ROS develop inflammatory response and lead to immune cells (CD45^+^, F4/80^+^) infiltration into the hepatic parenchyma. Inflammatory cytokines, produced by immune cells further activate HSCs and stimulate the production of collagen fibers and ECM deposition in the liver, leading to fibrogenesis [27].

TG excess in the serum of alb-myc^tg^ mice represent the mechanism by which FA are exported from the livers and delivered to adipose tissue for storage [28]. Hence, 36 weeks old transgenic animals developed considerably greater weight gain, BMI and e-WAT increase compare to control littermates on normal chow diet. Visceral adiposity and low-grade state of adipose tissue inflammation, assessed by presence of macrophage CLS in WAT, contribute to IR and hyperglycemia in transgenic mice [29,30]. High blood glucose level results in IR, and further contributes to metabolic disorders in the liver.

Tumor cells take in more glucose than normal to elevate aerobic glycolysis with the Warburg effect. Gene array data showed upregulation of PFKFB1-3 in transgenic liver, a key enzyme regulating glucose metabolism and promoting aerobic glycolysis. Hence, previous publications have shown that inhibitors targeting PFKFB3 have been found to suppress aerobic glycolysis, decrease glucose uptake, and induce cancer cell autophagy [31]. Moreover, consistently with previous publications transgenic expression of c-MYC also enhances the expression of mitochondrial GL and GLS for canonical glutaminolysis. Glutamine is the most abundant and nonessential amino acid that can be synthesized from glucose. Proliferative cells in many cancers, including some hepatocellular carcinomas, are glutamine addicted, and must induce GLS1 to increase glutamine catabolism and fuel their growth [32,33]. Interestingly, the hepatic GLS1 is also overexpressed in MAFLD patients [34].

Altogether, overexpression of c-MYC exclusively in the liver affects the whole-body metabolism and leads to the development of mild steatohepatitis/fibrosis in transgenic mice.

However, the development of MAFLD as well as further progression to HCC depends on multiple parallel factors, acting synergistically in genetically predisposed individuals. Thus, in order to evaluate a putative role of c-MYC in cancer initiation and progression we next intensified our model and applied high caloric WD to alb-myc^tg^ mice. In fact, this model will mimic the situation in human MAFLD, there combination of endogenous (e.g., oncogenes) and exogenous (food habits) factors synergistically contribute to the development of HCC.

As expected, c-MYC overexpression exacerbated steatosis, liver injury, liver inflammation, fibrosis and strong compensatory proliferation in transgenic mice fed with WD. Importantly, after 40 weeks of feeding all transgenic animals developed foci of tumors.

These findings are consistent with previously reported earlier HCC enhancement in c-MYC transgenic animals fed with either methionine choline-deficient diet (MCD) or choline-deficient and amino acid-defined diet (CDAA) [35]. Hence, we can conclude that metabolic changes induced by c-MYC promote hepatocarcinogenesis in a series of different murine MAFLD and HCC models.

Does the c-MYC transgenic mouse model reflect the clinical reality of human MAFLD-associated HCC? To answer this question, we analyzed human liver biopsy samples with MAFLD and MAFLD-associated HCC. Although a small cohort of 11 samples was used, we found an increased expression and prominent nuclear localization of the proto-oncogene c-MYC in patients with MAFLD and MAFLD-related HCC, highlighting its potentially critical role for the disease progression. Future studies with expanded sample size will needed to further confirm the role of c-MYC in human MAFLD development.

The limitations of a curative pharmacological treatment for human HCC clearly underline the need for the identification of novel target molecules. The MAFLD- HCC sequence suggest that specific inhibitors against metabolic signaling pathways may allow to interrupt the continuous transition from CLD to cancer.

Hence, in the current study we supplemented the chow diet with metformin and applied it to c-MYC transgenic animals. The use of metformin, the most commonly prescribed drug for DM2, was repeatedly associated with the decreased risk of the occurrence of various types of cancers, especially of pancreas and colon and HCC. This observation was also confirmed by the results of numerous meta-analyses that confirmed that metformin reduces cancer incidence by 30–50% [36]. Several previous publications reported MYC inhibition by metformin [37,38]. In the current study we could not detect the direct inhibition of c-MYC in the liver of transgenic mice by metformin-enriched diet. Hence we found that metformin had a strong inhibitory effects on SREBP1 expression in alb-myc^tg^ animals. Due to the lipogenic inhibition in the liver, alb-myc^tg^ mice treated with metformin diet were resistant to obesity, had modestly improved glucose and lipid metabolic parameters, and exhibited a marked lowering of liver steatosis and overall reduced collagen accumulation.

Our findings are consistent with previous reports showing the interaction of c-MYC with SREBP1 [39,40]. Indeed, as highly proliferating cancer cells require enhanced lipid production, c-MYC induces SREBP and together they collaborate to activate fatty acid (FA) synthesis and drive FA elongation from glucose and glutamine [41,42]. These findings point toward the promise of blocking c-MYC induced de novo lipogenesis as a general therapeutic strategy to combat common MYC-driven cancers. Hence in spite of remarkable improvement of steatohepatitis in alb-myc^tg^ mice treated with metformin, we could detect only mild changes in hepatic proliferation. Therefore, future studies will need to address the intriguing possibility of metformin to influence MYC-mediated cell growth.

In summary, phenotypical changes in the livers derived by c-MYC oncogene closely recapitulate the human MAFLD, albeit in a more telescope time frame. Hence, the evaluation of the molecular mechanisms underlying the associations between new genetic risk factors and progressive liver disease could be useful for the identification of novel therapeutic targets for MAFLD. Importantly, the detection of such preexisting risk factors should be used to identify the “high risk” MAFLD patients in whom preventive measures such as regular individual counselling and screening must be undertaken [41]. In fact, life expectancy of patients with HCC depends on the stage of the cancer at diagnosis. In advanced stages, some months are expected; however, if diagnosed in early stages, curative treatments such as surgical resection, liver transplant and local ablation can improve the survival of the patients and a 5-year survival rate can be reached. A very recent study demonstrated that the shorter survival of MAFLD-HCC patients, compared to HCV-HCC patients, is mainly due to the late diagnosis and greater tumor burden and not because HCC in MAFLD is more aggressive [42].

## 5. Conclusions

The current study, based on a clear and novel phenotype of transgenic mice, as well as interesting findings in patients, makes c-MYC an attractive target which can help to guide physicians towards preventative strategies and individualized therapy that can improve clinical outcome of MAFLD-related HCC.

## Figures and Tables

**Figure 1 cancers-14-00192-f001:**
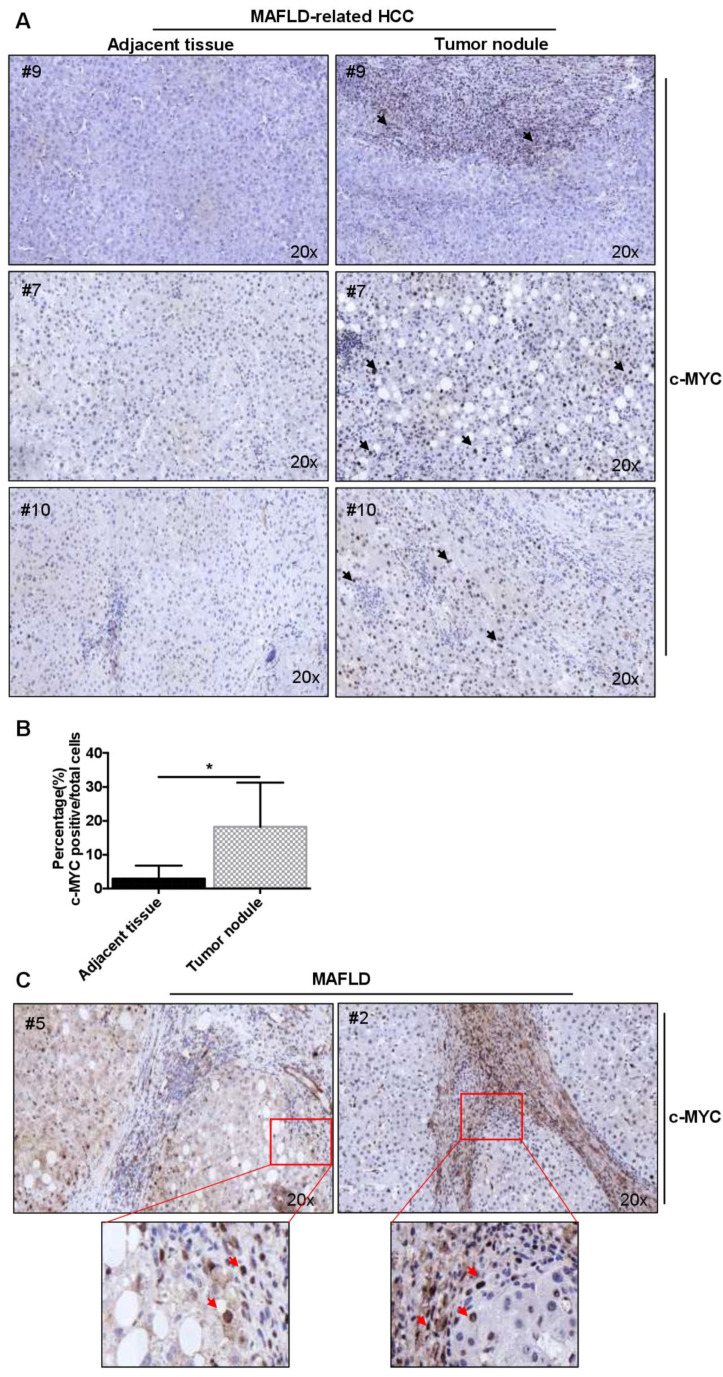
High expression of c-MYC in patients with MAFLD and MAFLD-related HCC. (**A**) c-MYC nuclear expression inside tumor nodules and tumor-adjacent tissues in patients with MAFLD-related HCC (20× magnification). Black arrows indicate cells with positive nuclear c-MYC expression (**B**) Quantification of % c-MYC positive cells inside tumor nodules and tumor-adjacent tissues in patients with MAFLD-related HCC (*n* = 4). (**C**) c-MYC protein expression in MAFLD patients with advanced liver fibrosis (F3–F4), (20× magnification). Red arrows indicate cells with positive nuclear c-MYC expression. Data are expressed as the mean ± SD, * = *p* < 0.05, inside tumor nodules vs. tumor-adjacent tissues.

**Figure 2 cancers-14-00192-f002:**
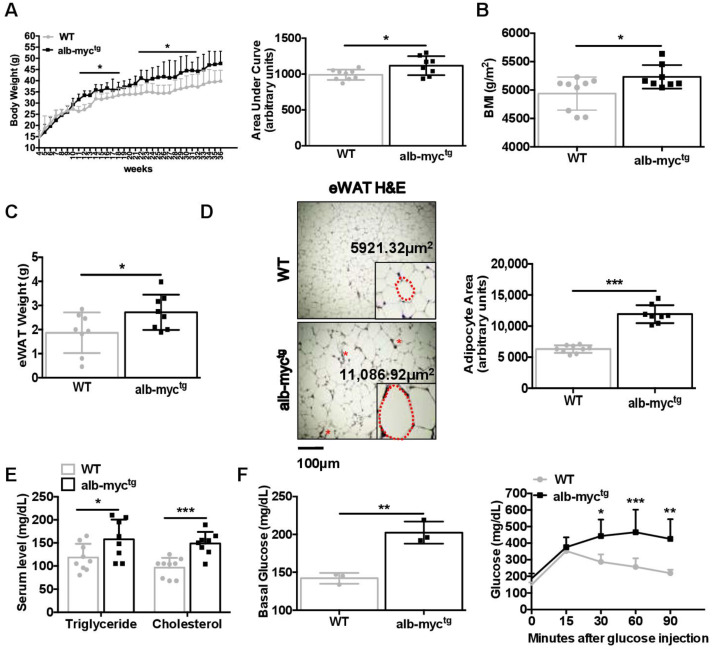
Metabolic profile of chow diet-fed alb-myc^tg^ mice at the age of 36 weeks. (**A**) Left: Growth curve demonstrating body weight gain, assessed at 7 days’ interval for each group. Right: The histogram represents the incremental area under the respective grow curve (*n* = 8–9). (**B**) Body mass index (BMI) (g/m^2^) (*n* = 8–9). (**C**) Weight of epididymal white adipose tissue (eWAT) (*n* = 8). (**D**) Size of eWAT cells by H&E staining. Scale bar = 100 µm. Size of individual cells from both groups was measured and average cell size (µm^2^) was quantified (*n* = 8–9). Asterisks indicate CLS. The dotted red line marks the individual adipocyte. (**E**) Levels of cholesterol and TG in serum (*n* = 8–9). (**F**) Basal glucose level in blood after 6 h fasting; Blood glucose concentration curve during intraperitoneal glucose tolerance test (*n* = 3). Data are expressed as the mean ± SD, * = *p* < 0.05; ** = *p* < 0.01, *** = *p* < 0.001, alb-myc^tg^ mice vs. WT controls.

**Figure 3 cancers-14-00192-f003:**
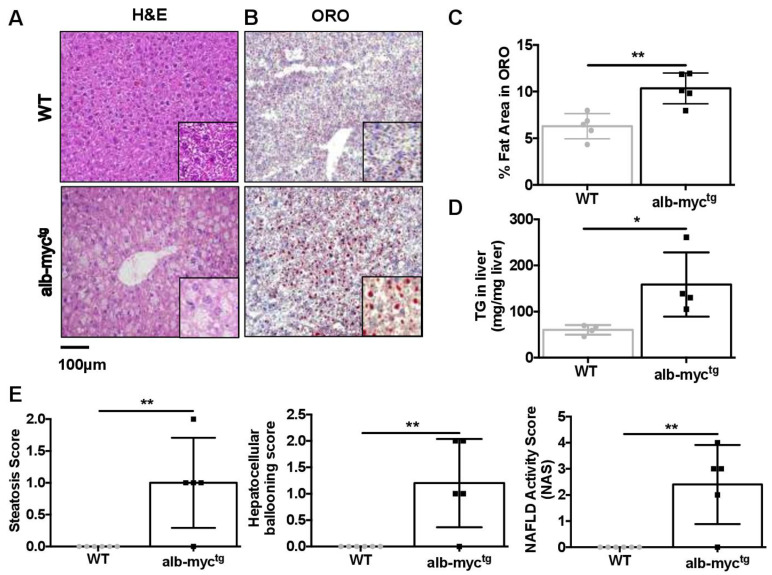
Steatotic changes in the liver of transgenic mice. (**A**) Representative liver H&E staining demonstrating distended hepatocytes with foamy appearing cytoplasm and small lipid vesicles. Scale bar = 100 µm (*n* = 8–9). (**B**) Illustrative ORO staining. Scale bar = 100 µm. (**C**) Quantitative analysis of ORO-stained area (*n* = 5). (**D**) Direct TG quantification (*n* = 4). (**E**) Histology score for steatosis, hepatocyte ballooning, NAFLD activity score (NAS) (*n* = 5–6). Data are expressed as the mean ± SD, * = *p* < 0.05; ** = *p* < 0.01, alb-myc^tg^ mice vs. WT controls.

**Figure 4 cancers-14-00192-f004:**
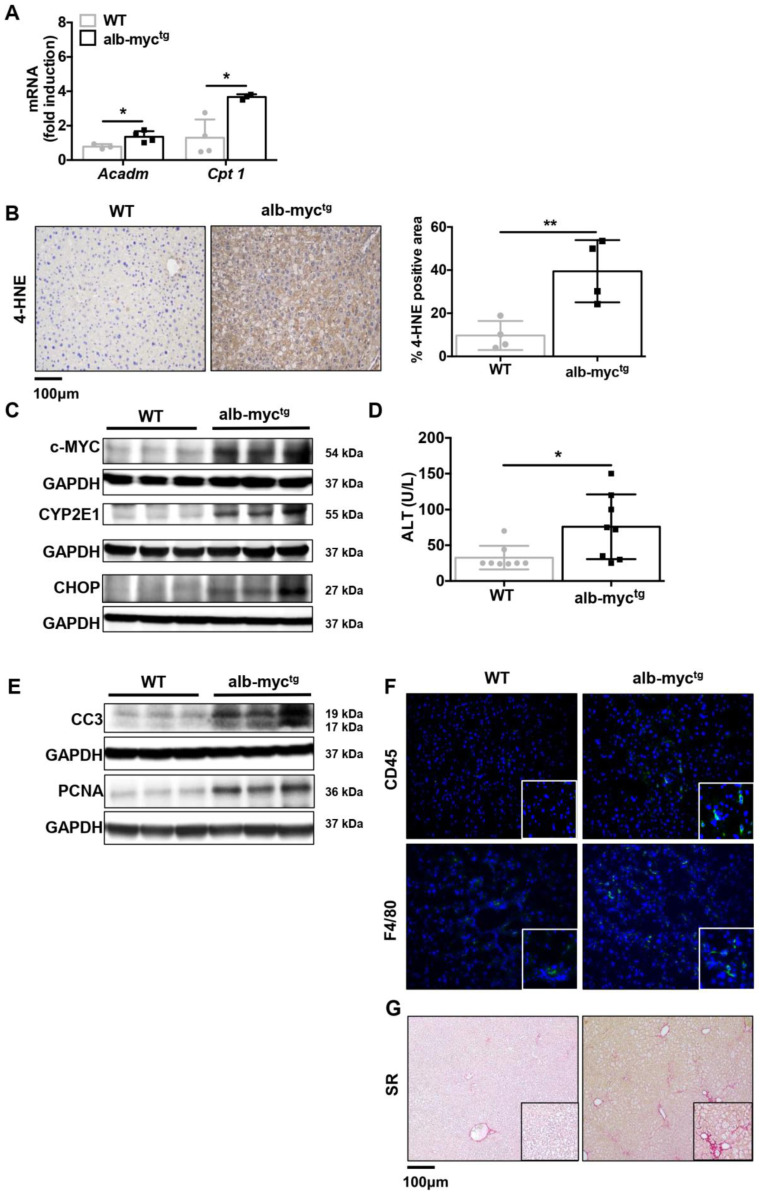
Transgenic overexpression of c-MYC in the livers leads to induction of ER stress, apoptosis, inflammation and fibrosis. (**A**) qPCR analysis of hepatic mRNA expression of Acadm and Cpt1 genes (*n* = 6). (**B**) 4-HNE expression (brown) of representative livers was evaluated by immunohistochemistry. Scale bar = 100 µm. The histogram represents the 4-HNE staining intensity (% square microns) (*n* = 4). (**C**) Immunoblot analysis of c-MYC, CYP2E1, CHOP (*n* = 3). Uncropped blots are presented in Appendix A. (**D**) Levels of ALT in serum (*n* = 8–9). (**E**) Protein expression of CC3 and PCNA (*n* = 3). GAPDH was used as a control for protein loading). Uncropped blots are presented in Appendix A. (**F**) Representative IF images of hepatic inflammation (CD45 and F4/80 positive cells are stained in green). (**G**) Sirius red (SR) staining of liver paraffin sections showing accelerated onset of fibrosis formation in alb-myc^tg^. * = *p* < 0.05; ** = *p* < 0.01.

**Figure 5 cancers-14-00192-f005:**
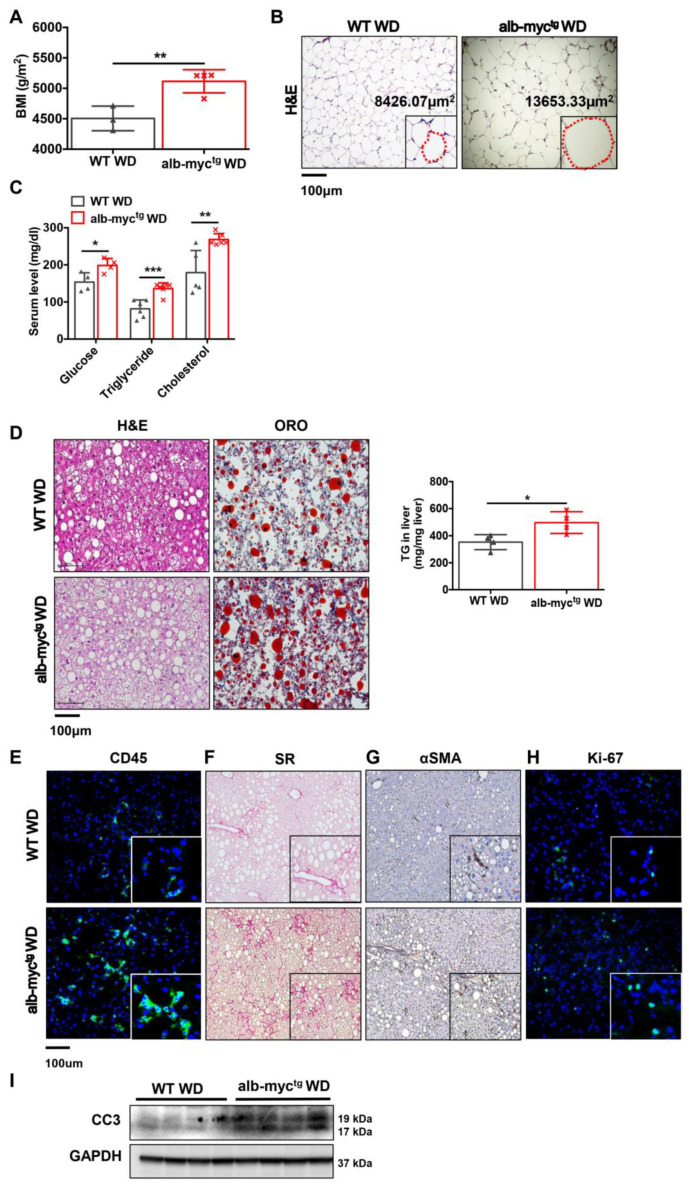
WD feeding accelerates liver injury in alb-myct^g^ mice. WT (*n* = 6) and alb-myc^tg^ (*n* = 7) mice fed for 24 weeks with WD. (**A**) Body mass index (BMI) (g/m^2^) of mice was calculated at the age of 36 weeks (*n* = 3–4). (**B**) H&E staining of epididymal white adipose tissue (eWAT). The dotted red line marks the individual adipocyte. Scale bar = 100 µm. (**C**) Fasting blood glucose, levels of cholesterol and TG in serum (*n* = 6–8). (**D**) Representative liver sections stained with H&E and ORO are shown. Scale bar = 100 µm. Histogram showing TG quantification of mice fed WD (*n* = 4). (**E**) Representative images of hepatic inflammation (CD45 positive cells stained in green). (**F**,**G**) Liver fibrosis (SR and αSMA IHC staining). (**H**) Hepatic proliferation (Ki-67 positive cells in green). Scale bar = 100 µm. (**I**) Protein expression of cleaved caspase 3, GAPDH used as loading control (*n* = 3). Uncropped blots are presented in Appendix A. Data are expressed as the mean ± SD, * = *p* < 0.05, ** = *p* < 0.01, *** = *p* < 0.001, alb-myc^tg^ mice fed with WD compared to WT fed with WD.

**Figure 6 cancers-14-00192-f006:**
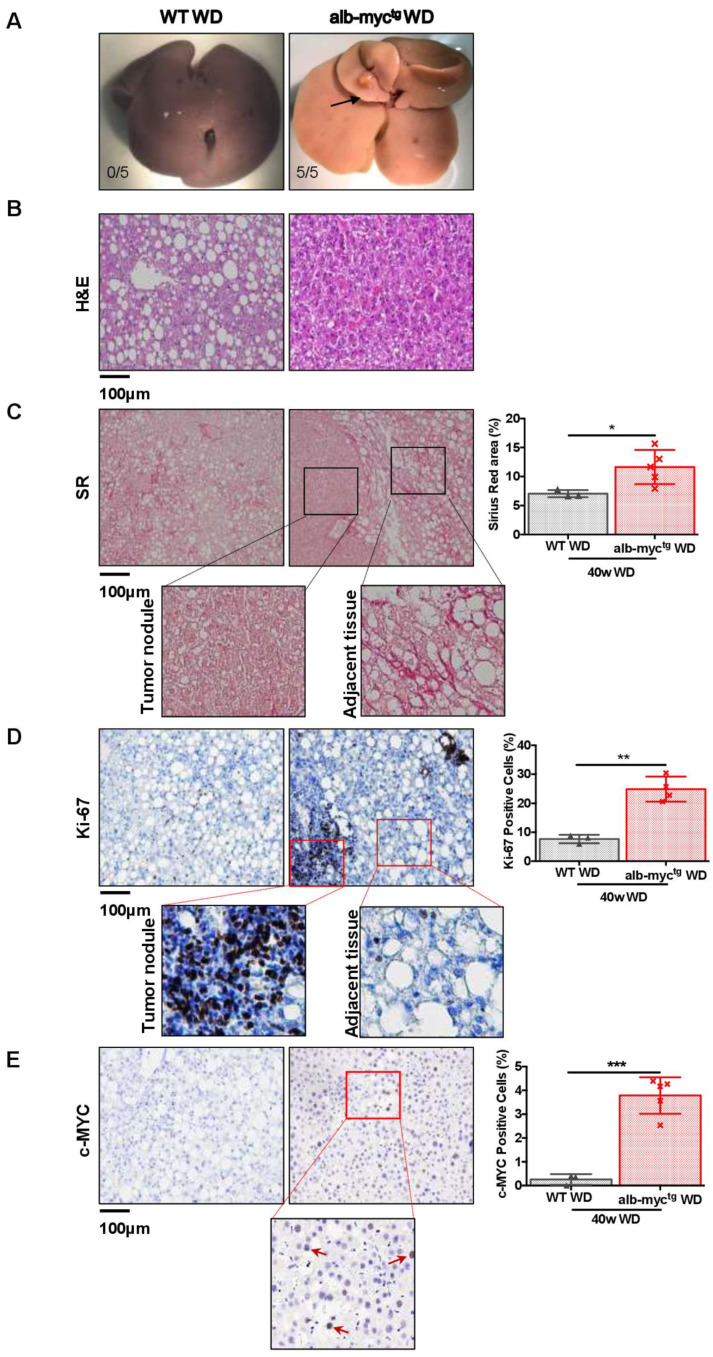
WD feeding accelerates the initiation of MAFLD associated tumorigenesis in alb-myc^tg^ mice. WT (*n* = 3) and alb-myc^tg^ (*n* = 5) mice fed 40 weeks with WD. (**A**) Gross liver. Numbers represent tumor incidence. (**B**) Representative liver sections stained with H&E. (**C**) Representative liver sections stained with SR. Quantification of % SR positive area (*n* = 3–5) Scale bar = 100 µm. (**D**) Ki-67 immunostaining of paraffin sections showing increased cell proliferation (brown) of hepatocytes in alb-myc^tg^ mice liver. Quantification of Ki-67 positive cells (*n* = 3–5). Scale bar = 100 µm. (**E**) c-MYC IHC staining of paraffin sections. Quantification of c-MYC positive cells (*n* = 3–5). Scale bar = 100 µm. * = *p* < 0.05, ** = *p* < 0.01, *** = *p* < 0.001.

**Figure 7 cancers-14-00192-f007:**
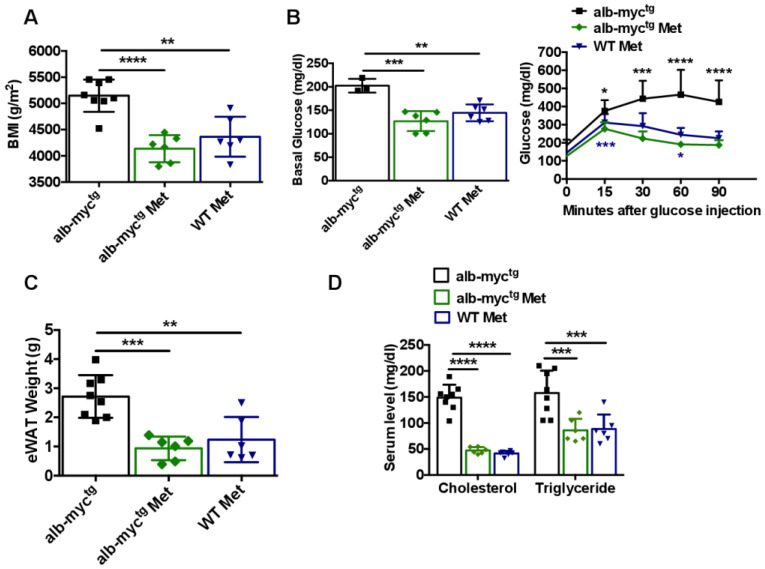
Improved hepatic phenotype in alb-myc^tg^ mice after metformin treatment. alb-myc^tg^ mice (*n* = 6) and WT mice (*n* = 6) fed 20 weeks received metformin enriched chow diet, alb-myc^tg^ (*n* = 8) received chow diet; all animals were sacrificed at the age of 36 weeks. (**A**) Body mass index (BMI) (g/m^2^) of mice at the age of 36 weeks (*n* = 3–6). (**B**) Basal glucose level after 6 h fasting. Blood glucose concentration curve during intraperitoneal GTT. (*n* = 3–6). (**C**) eWAT weight (*n* = 6–8). (**D**) Serum levels of cholesterol and TG in serum (*n* = 6–8). Representative liver sections stained with H&E (**E**), ORO (**F**), SR (**G**) and Ki-67 (**H**) stainings. Scale bar = 100 µm. (**I**) Protein expression of SREBP-1, GAPDH used as loading control (*n* = 3). Uncropped blots are presented in Appendix A. Data are expressed as the mean ± SD, * = *p* < 0.05, ** = *p* < 0.01, *** = *p* < 0.001, **** = *p* < 0.0001, alb-myc^tg^ mice fed metformin enriched chow diet vs. alb-myc^tg^ mice fed chow diet; ** = *p* < 0.01, *** = *p* < 0.001, **** = *p* < 0.0001, WT mice fed metformin enriched chow diet vs. alb-myc^tg^ mice fed chow diet.

## Data Availability

Not applicable.

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
