# Peer review of "A Shortcut from Metabolic-Associated Fatty Liver Disease (MAFLD) to Hepatocellular Carcinoma (HCC): c-MYC a Promising Target for Preventative Strategies and Individualized Therapy"

_cancers, 2021, doi:10.3390/cancers14010192_

Round 1

Reviewer 1 Report

Guo, Estévez-Vázquez, Benedé-Ubieto, and colleagues investigated the functional roles of c-Myc during the progression from metabolic-associated fatty liver disease to hepatocellular carcinoma. They mainly employed the c-MYC transgenic mice and studied the effects of WD feeding and Metformin on MAFLD-associated changes. Data are carefully analyzed and clearly presented. A minor revision is needed to solve the following questions.

Comments:

  1. When authors analyzed the genetic alterations of c-MYC in a comprehensive dataset, if possible, it would be important to identify the percentage of people with WD diets.
  2. In Figure 4C, is the upper band the activated CHOP? Please clarify in the legend.
  3. It is critical to assess the expression level of c-Myc after metformin treatment in Figure 7.
  4. Staining of the adjacent tissues is important in several figures, eg Figures 6 and 7E.
  5. Proofreading is needed because there are some typos, missing punctuation, and format errors (blank regions at the end of some pages).

Author Response

REVIEWER#1

Guo, Estévez-Vázquez, Benedé-Ubieto, and colleagues investigated the functional roles of c-Myc during the progression from metabolic-associated fatty liver disease to hepatocellular carcinoma. They mainly employed the c-MYC transgenic mice and studied the effects of WD feeding and Metformin on MAFLD-associated changes. Data are carefully analyzed and clearly presented. A minor revision is needed to solve the following questions.

We thank a Reviewer for a positive feedback.

Comments:

1. When authors analyzed the genetic alterations of c-MYC in a comprehensive dataset, if possible, it would be important to identify the percentage of people with WD diets.

Thank you for this interesting suggestion. For the manuscript we used data from PanCancer Atlas. We extracted the information concerning the number of patients with genetic alteration, mutation spectrum, diagnosis age and overall survival. Unfortunately, it was impossible to track the etiology of HCC with c-MYC alteration.

2.In Figure 4C, is the upper band the activated CHOP? Please clarify in the legend.

Thank you, certainly the upper band is specific. We added the indicating arrow to Fig.4C.

 3.It is critical to assess the expression level of c-Myc after metformin treatment in Figure 7.

Thank you for rising this important question. We believe, that positive phenotypical changes in transgenic mice fed with metformin diet are not a consequence of the direct c-MYC inhibition (mRNA level of c-MYC has not been significantly changed after treatment and was still high, Suppl. Fig 12E). However, we found significant downregulation of the downstream target of c-MYC – lipogenic regulator SREBP1 (Fig. 7I). Our findings are consistent with previous reports showing the crucial interaction of c-MYC with SREBP1. Our results indicate that blocking c-MYC induced de novo lipogenesis can be a promising therapeutic strategy to combat MYC driven MAFLD and MAFLD associated HCC.

 4. Staining of the adjacent tissues is important in several figures, eg Figures 6 and 7E.

Thank you. We have accordingly modified Fig. 6.

5. Proofreading is needed because there are some typos, missing punctuation, and format errors (blank regions at the end of some pages).

Thank you, we have proofread the manuscript.

Reviewer 2 Report

In the manuscript titled “A shortcut from Metabolic-Associated Fatty Liver Disease (MAFLD) to hepatocellular carcinoma (HCC): c-MYC a promising theranostic target” Guo et al. have studied the impact of c-MYC on the progression of MAFLD and MAFLD-associated HCC. The authors have observed the elevated level of c-MYC in biopsies of patients with MAFLD and MAFLD-associated HCC in the immunohistochemistry study. The 36 weeks study performed on transgenic mice with overexpressed c-MYC in hepatocytes (alb-myctg) revealed that these mice are more prone to develop obesity, metabolic syndrome (MS), abnormal accumulation of lipids in the liver compared to control mice. The authors have also determined that the liver injury and inflammation in alb-myctg mice are linked to upregulation of ALT, ER-stress response, increased ROS production and collagen deposition. Overall, the study indicates a strong link between c-MYC and MAFLD-associated HCC. I believe that these findings will help in future developments on strategies and therapies to improve the MAFLD-related HCC treatment. I recommend the publication of this manuscript in Cancers in its current form.

Author Response

Thank you very much for such a positive feedback.

Reviewer 3 Report

This manuscript (ms) is promising but needs improvements.

Title: The words ‘shortcut’ and ‘theranostic’ are not appropriate. Redesign the title.

This definition of theranostic  explains why this term cannot be used in this manuscript.

Theranostics is a combination of the terms therapeutics and diagnostics. Theranostics is the term used to describe the combination of using one radioactive drug to identify (diagnose) and a second radioactive drug to deliver therapy 

  1. Is the effect of cMyc related to ageing or to a need for a period of time needed for cMyc overexpression , just like many other liver insults, to cause outcomes? Mice given DEN at birth show tumours at 9-12 months of age but this outcome has not been ascribed to ageing. Ageing effects in mice occur at 18 or more months of age, not within the first year of life.

As the cMyc overexpression was not compared between young and old mice, the claim that the observed outcomes are a synergy of cMyc and ageing is not supported.

  1. The mouse model is of cMyc overexpression, so the parallel human data must be cMyc overexpression. However, the text of this manuscript often muddles cMyc overexpression with expression of mutant forms of cMyc. Clarity must be increased. For example, line 117 on page 3 vaguely mentions ‘alterations, but only cMyc variants that cause cMyc overexpression are relevant.
  2. cMyc and MYC and cMYC are synonyms. Protein names regular font whereas gene name should be in italics. Human genes are written with all capitals, mouse genes have capital as first letter. For example, Suppl Table 7 has a mix of styles: does this mean that some primers were for human genes and some for mouse genes?
  3. Abstract: Should have greater clarity. ’52 weeks, 20%...’ is not results data from the current manuscript, it is in Ref 15; the only HCC data in this ms is in Fig 6. alb-myctg must be defined. ‘theranostic’ must be removed.
  4. The final two paragraphs of introduction need re-writing for clarity and precision. For example, ‘unravelled’ is not meaningful. For example, this ms does not focus on ‘end-stage’ HCC.
  5. Liver damage of any kind usually impairs liver functions , which is primarily in metabolism. So, a causal link is needed and is lacking between cMyc and a metabolic pathway. The metformin experiment supports the idea that such a link exists. But the link is not identified. This should be acknowledged in discussing this data.
  6. Only 6 human MAFLD patients were analysed, and 2 of them did not have steatosis. Only 4 MAFLD/HCC patients were studied. Either pull back on interpretation of data from these patients, or add more patients. Page 3 line 139 needs clear precise explanation of the ‘histopathological’ criteria.
  7. Ref 10 is referred to for the cMyc mouse. Be clear about what is already known about this mouse from previous work and what data in this ms is repeated experiments.
  8. Page 4 live 167: the cMyc variants studied here must only be variants known to cause overexpression. Other variants are not relevant to the current ms, which concerns cMyc overexpression. ‘mainly amplifications’ is imprecise and needs correction.
  9. Fig 1. There appears to be nuclear immunostain, but it is difficult to see. 2-colour fluorescence, showing each colour in turn, or 1-colour IHC with brown stain only, without the blue stain overlay, would be much more helpful for seeing nuclear stain. Why would nuclear cMyc mainly be near regions of fibrosis? This observation deserves discussion.
  10. Fig 2A is inadequate. BW was measured each 7 days, so each data point should be graphed rather than a wiggly line of unknown provenance. For example, a line joining means, as done in Fig2F, would be appropriate. The number of mice for Fig2A must be stated.
  11. Fig 2D needs to also have added a fig of IHC of a macrophage marker, such as F4/80 or CD68 to prove that what is claimed to be macrophages is macrophages. Define ‘CLS’. Explain the red dots.
  12. All figures that show BMI data are confusing because the abscissa is labelled ‘(g/m2)’ but in the text BMI is claimed to be ‘m2/BW’.
  13. Fig 3C. AU is not acceptable. Surely ‘positive area’ is a percentage of total area?
  14. Fig 4 F is unconvincing. These 3 parameters must be quantified.
  15. Fig 5. As for Fig 4, immunostained sections parameters [ORO, CD45, SR, aSMA, Ki67] must be quantified and the graphs shown here. WAT should have a macrophage stain. Again, and in Fig 7; m2/BW or g/m2? In addition, this fig lacks a chart of BW over time; this needs to be added.
  16. Define DM2.
  17. Fig 6: quantify all the parameters that are analysed in mouse livers.
  18. Discussion exhibits a tendancy to over-reach in claims made. Please carefully reconsider claims to ensure that all are well supported. For example, line 430, ‘perfectly mimic’. For example, there is no evidence of metformin influencing tumour or pre-tumourous cell proliferation [line 465-8].
  19. Discussion and Conclusion: remove ‘theranostic’

Author Response

This manuscript (ms) is promising but needs improvements.

Title: The words ‘shortcut’ and ‘theranostic’ are not appropriate. Redesign the title.

This definition of theranostic explains why this term cannot be used in this manuscript.Theranostics is a combination of the terms therapeutics and diagnostics. Theranostics is the term used to describe the combination of using one radioactive drug to identify (diagnose) and a second radioactive drug to deliver therapy 

Thank you for this insightful comment. In the revised version, we changed the title to ¨A shortcut from metabolic-associated fatty liver disease (MAFLD) to hepatocellular carcinoma (HCC): c-MYC, a promising theranostic target for preventive strategies and individualized therapy ¨

Major comments:

  1. Is the effect of cMyc related to ageing or to a need for a period of time needed for cMyc overexpression, just like many other liver insults, to cause outcomes? Mice given DEN at birth show tumours at 9-12 months of age but this outcome has not been ascribed to ageing. Ageing effects in mice occur at 18 or more months of age, not within the first year of life. As the cMyc overexpression was not compared between young and old mice, the claim that the observed outcomes are a synergy of cMyc and ageing is not supported.

Thank you for this interesting point. We believe that a certain period of time is needed for c-Myc overexpression to cause metabolic syndrome and MAFLD changes in the liver. For example, as it shown in Fig 1A (for Reviewers only) the first body weight changes are detectable in 10 week-old transgenic mice. As the mice are getting older the changes in body weight became more pronounced and accompanied by corresponding changes in the liver. Consistently, we could not detect significant fat accumulation in the liver tissue at 26 week-ld chow fed transgenic animals, but at 36 weeks the changes are already significant compared to control mice (Fig. 1 (for reviewers only) and revised manuscript Fig 2 B-C). The corresponding lines have been added to the Results and Discussion sections.

Fig.1 (for Reviewers only) Representative ORO staining demonstrating no significant difference in the fat accumulation in the liver of 26 weeks old alb-myctg mice (n=3-4)

  1. The mouse model is of cMyc overexpression, so the parallel human data must be cMyc overexpression. However, the text of this manuscript often muddles cMyc overexpression with expression of mutant forms of cMyc. Clarity must be increased. For example, line 117 on page 3 vaguely mentions ‘alterations, but only cMyc variants that cause cMyc overexpression are relevant.

Thank you for this comment. We would like to point out that c-MYC can be up-regulated in cancer on many levels including increase in copy number (referred to as amplification), constitutive up-regulation of the MYC promoter (leading to transcriptional over-expression) or increase of c-myc protein stability as reviewed e.g. by Kalkat et al1.Accordingly, both c-MYC amplification (as found in many human cancer entities within the human genome atlas) as well as c-MYC over-expression eventually results in enhanced c-MYC transcripts. The only difference is that in the case of amplifications the transcripts are derived from different copies of the MYC gene, while in true over-expression the enhanced transcription is the result of a single constitutively hyper-active promoter. However, the consequence is similar: elevated level of c-MYC protein. In conclusion, the data on c-myc amplification in the human genome atlas are definitely relevant for our manuscript and strengthen our own findings. We hope that these explanations were able to adequately answer the Reviewer's question.

  1. cMyc and MYC and cMYC are synonyms. Protein names regular font whereas gene name should be in italics. Human genes are written with all capitals, mouse genes have capital as first letter. For example, Suppl Table 7 has a mix of styles: does this mean that some primers were for human genes and some for mouse genes?

Thank you, we apologize for the inappropriate nomenclature. The Suppl. Tables 7 and 8 has been changed accordingly.

  1. Abstract: Should have greater clarity. ’52 weeks, 20%...’ is not results data from the current manuscript, it is in Ref 15; the only HCC data in this ms is in Fig 6. alb-myctg must be defined. ‘theranostic’ must be removed.

We apologized for the lack of clarity. The HCC data in 52 weeks old mice (Suppl Fig. 10) have been collected during the current project and are in total agreement with previous publication Ref 16. We modified the final paragraphs of the Result section 3.3 accordingly. The word “theranostic” has been removed from the manuscript.

  1. The final two paragraphs of introduction need re-writing for clarity and precision. For example, ‘unravelled’ is not meaningful. For example, this ms does not focus on ‘end-stage’ HCC.

Thank you for this suggestion, we completely changed the two final paragraphs of the introduction

  1. Liver damage of any kind usually impairs liver functions, which is primarily in metabolism. So, a causal link is needed and is lacking between cMyc and a metabolic pathway. The metformin experiment supports the idea that such a link exists. But the link is not identified. This should be acknowledged in discussing this data.

We thank the Reviewer for this suggestion. We found that overexpression of SREBP1 in alb-myctg mice was strongly diminished by metformin treatment. This information is now added to the revised manuscript (Results 3.5 and Discussion)

  1. Only 6 human MAFLD patients were analysed, and 2 of them did not have steatosis. Only 4 MAFLD/HCC patients were studied. Either pull back on interpretation of data from these patients, or add more patients. Page 3 line 139 needs clear precise explanation of the ‘histopathological’ criteria.

Thank you for pointing this out. Yes, we agree that the small cohort of patients is a serious limitation of this study. We added the required information to the discussion.

All HCC samples used in this study were analysed by experience pathologist and had trabecular-resembling and solid growth patterns, this information has also been added to the manuscript.

  1. Ref 10 is referred to for the cMyc mouse. Be clear about what is already known about this mouse from previous work and what data in this ms is repeated experiments.

Thank you, we modified the Introduction and Discussion, accordingly.

  1. Page 4 live 167: the cMyc variants studied here must only be variants known to cause overexpression. Other variants are not relevant to the current ms, which concerns cMyc overexpression. ‘mainly amplifications’ is imprecise and needs correction.

Thank you, as we pointed in the reply to the comment 2, both c-MYC amplification as well as c-MYC over-expression eventually results in enhanced c-MYC transcripts; therefore, we believe that the data on c-MYC amplification in the human genome atlas are relevant for our manuscript.

  1. Fig 1. There appears to be nuclear immunostain, but it is difficult to see. 2-colour fluorescence, showing each colour in turn, or 1-colour IHC with brown stain only, without the blue stain overlay, would be much more helpful for seeing nuclear stain. Why would nuclear cMyc mainly be near regions of fibrosis? This observation deserves discussion.

Thank you, we added corresponding IHC images with only DAB (nuclear staining) and with only hematoxylin (Suppl. Fig 2 in the revised manuscript).

Yes, we agree, nuclear c-MYC expression in hepatocytes is located near the region of fibrosis. Similar findings have been reported by us in a previous paper2. We believe that in early stages c-MYC acts predominantly in hepatocytes. The up-regulation of c-MYC in hepatocytes either due to genetic gene amplification or as a response to inflammatory liver injury results in moderate hepatocyte apoptosis, enhanced hepatocyte proliferation and aberrant expression of PDGF-B. Close physical vicinity of dying or PDGF-expressing hepatocytes with resident quiescent HSC can synergistically lead to HSC pre-activation and moderate trans-differentiation into myofibroblasts with subsequent mild collagen production3.

  1. Fig 2A is inadequate. BW was measured each 7 days, so each data point should be graphed rather than a wiggly line of unknown provenance. For example, a line joining means, as done in Fig3F, would be appropriate. The number of mice for Fig3A must be stated.

Thank you, in the revised version, we changed the diagram accordingly and we stated the number of mice in the figure legend (n=8-9). We also modified the diagram 12A in Supplementary Materials.

  1. Fig 2D needs to also have added a fig of IHC of a macrophage marker, such as F4/80 or CD68 to prove that what is claimed to be macrophages is macrophages. Define ‘CLS’. Explain the red dots.

Thank you for this valuable comment. We performed the IF F4/80 staining in WAT tissue, the corresponding pictures have been added to Suppl. Fig 3C. The line with red dots marks the individual adipocytes and simplifies the size comparison for the readers. The definition of CLS has been added to the text.

  1. All figures that show BMI data are confusing because the abscissa is labelled ‘(g/m2)’ but in the text BMI is claimed to be ‘m2/BW’.

We apologize for this mistake. BMI is weight (in g) divided by height squared (in meters). The corresponding changes have been done in the manuscript.

  1. Fig 3C. AU is not acceptable. Surely ‘positive area’ is a percentage of total area?

Thank you. The positive area is a percentage. We changed the labelling of the Y axis to “% Fat area in ORO” (Fig 3C, Suppl. 11D).

  1. Fig 4 F is unconvincing. These 3 parameters must be quantified.

Thank you. The quantifications for Fig. 4F are represented as Suppl. fig. 7B-C and for Fig. 4G as Suppl. Fig. 7D.

  1. Fig 5. As for Fig 4, immunostained sections parameters [ORO, CD45, SR, aSMA, Ki67] must be quantified and the graphs shown here. WAT should have a macrophage stain. Again, and in Fig 7; m2/BW or g/m2? In addition, this fig lacks a chart of BW over time; this needs to be added.

Thank you, the figures have been modified according your suggestions:

  • Quantification for 5D (ORO) is Suppl. Fig. 11D
  • Quantification for 5E (CD45) is Suppl. Fig. 11F
  • Quantification for 5F (SR) is Suppl. Fig. 11G
  • Quantification for 5G (aSMA) is Suppl. Fig. 11H
  • Quantification for 5H Ki67is Suppl. Fig. 11J
  • We apologized for a wrong labeling in Fig 7A. Now it is g/m2
  • The chart of BW over time is added to Suppl Fig 11 A
  1. Define DM2.

Thanks, the definition of DM2 is now included in the Introduction.

  1. Fig 6: quantify all the parameters that are analysed in mouse livers.

Thank you, the quantifications have been added to Fig. 6

  1. Discussion exhibits a tendancy to over-reach in claims made. Please carefully reconsider claims to ensure that all are well supported. For example, line 430, ‘perfectly mimic’. For example, there is no evidence of metformin influencing tumour or pre-tumourous cell proliferation [line 465-8].

Thank you, we toned-down the Discussion.

  1. Discussion and Conclusion: remove ‘theranostic’

Thank you. It has been removed.

References

1          Kalkat, M. et al. MYC Deregulation in Primary Human Cancers. Genes (Basel) 8, doi:10.3390/genes8060151 (2017).

2          Nevzorova, Y. A. et al. Enhanced expression of c-myc in hepatocytes promotes initiation and progression of alcoholic liver disease. J Hepatol 64, 628-640, doi:10.1016/j.jhep.2015.11.005 (2016).

3          Nevzorova, Y. A. et al. Overexpression of c-myc in hepatocytes promotes activation of hepatic stellate cells and facilitates the onset of liver fibrosis. Biochim Biophys Acta 1832, 1765-1775, doi:10.1016/j.bbadis.2013.06.001 (2013).

Round 2

Reviewer 3 Report

Thank you. I agree with all the arguments made in response to reviewer comments and the modifications to the manuscript made by the authors.